# Genomic analysis of focal nodular hyperplasia with associated hepatocellular carcinoma unveils its malignant potential: a case report

Caner Ercan [1,2,10], Mairene Coto-Llerena[1,2,10], John Gallon [1,10], Lana Fourie[1,3], Mattia Marinucci [1], Gabriel F. Hess [1,3], Jürg Vosbeck[2], Stephanie Taha-Mehlitz[3], Tuyana Boldanova[3,4], Marie-Anne Meier[4], Alexandar Tzankov [2], Matthias S. Matter [2], Martin H. K. Hoffmann[5], Luca Di Tommaso[6,7], Markus von Flüe[3], Charlotte K. Y. Ng [8,9], Markus H. Heim [3,4], Savas D. Soysal[3], Luigi M. Terracciano[6,7], Otto Kollmar[3] & Salvatore Piscuoglio [1,2✉]

## Abstract

**Background** Focal nodular hyperplasia (FNH) is typically considered a benign tumor of the liver without malignant potential. The co-occurrence of FNH and hepatocellular carcinoma (HCC) has been reported in rare cases. In this study we sought to investigate the clonal relationship between these lesions in a patient with FNH-HCC co-occurrence.

**Methods** A 74-year-old female patient underwent liver tumor resection. The resected nodule was subjected to histologic analyses using hematoxylin and eosin stain and immunohistochemistry. DNA extracted from microdissected FNH and HCC regions was subjected to whole exome sequencing. Clonality analysis were performed using PyClone.

**Results** Histologic analysis reveals that the nodule consists of an FNH and two adjoining HCC components with distinct histopathological features. Immunophenotypic characterization and genomic analyses suggest that the FNH is clonally related to the HCC components, and is composed of multiple clones at diagnosis, that are likely to have progressed to HCC through clonal selection and/or the acquisition of additional genetic events.

**Conclusion** To the best of our knowledge, our work is the first study showing a clonal relationship between FNH and HCC. We show that FNH may possess the capability to undergo malignant transformation and to progress to HCC in very rare cases.

## Plain language summary

Focal nodular hyperplasia (FNH) is a lesion resulting from the abnormal growth of liver cells. It is typically considered a benign tumor that does not become malignant. In rare cases, FNH may occur alongside malignant hepatocellular carcinoma (HCC). In these cases, it is not known whether the malignant HCC may derive from the benign FNH. In this study, we reported on the analysis of a 74-year-old female patient with co-occurring FNH and HCC. We found that the FNH and HCC lesions were in fact genetically related, suggesting that the FNH gave rise to the HCC lesions. Furthermore, we found multiple cell populations within the FHN lesion that may be precursors to the HCC lesions suggesting that, in rare cases, FNH may be capable of progressing to malignant HCC. These findings may help to refine the surveillance strategy for these lesions.

[1] Visceral Surgery and Precision Medicine Research Laboratory, Department of Biomedicine, University of Basel, Basel, Switzerland. [2] Institute of Medical Genetics and Pathology, University Hospital Basel, Basel, Switzerland. [3] Clarunis, University Center for Gastrointestinal and Liver Diseases, St. Clara Hospital and University Hospital Basel, Basel, Switzerland. [4] Hepatology Laboratory, Department of Biomedicine, University of Basel, Basel, Switzerland. [5] Department of Radiology, St. Claraspital, Basel, Switzerland. [6] Department of Pathology, Humanitas Clinical and Research Center, IRCCS, Rozzano, Milan, Italy. [7] Humanitas University, Department of Biomedical Sciences, Pieve Emanuele, Milan, Italy. [8] Department for BioMedical Research, University of Bern, Bern, Switzerland. [9] SIB Swiss Institute of Bioinformatics, Lausanne, Switzerland. [10] These authors contributed equally: Caner Ercan, Mairene Coto-Llerena, John Gallon. ✉email: s.piscuoglio@unibas.ch

Focal nodular hyperplasia (FNH) accounts for up to 8% of all liver tumors and is the second most common benign tumor in the liver[1]. The widely accepted theory is that FNH develops from a hyperplastic response to an increased local blood flow[2]. Besides the minimal risk of hemorrhage and rupture, FNH manifests as an indolent clinical disease that is mostly detected incidentally[2].

The development of hepatocellular carcinoma (HCC) is driven by progressively accumulating genetic, epigenetic, and micro-environmental alterations mostly in the background of chronic liver disease[3]. It is accepted that the multistep sequence of hepatocarcinogenesis includes progression through dysplastic lesions and hepatocellular adenomas[3]. However, this model does not explain the subset of HCCs neither associated with any background disease nor with defined precancerous lesions.

Co-occurrence of FNH and HCC has been reported in rare cases[4]. Previous studies using comparative genomic hybridization or HUMARA clonality analysis of a limited number of genomic loci found that the alterations present in HCC could not be detected in synchronous FNHs[5–7]. However, no synchronous FNH and HCC compartments have been genomically characterized using comprehensive methods, so their clonal connection has not been properly assessed. Transcriptomic analysis of FNH has demonstrated only that two angiopoietin genes (ANGPT1, ANGPT2) have altered mRNA expression levels, without somatic gene mutations commonly observed in HCC[8].

FNH is rarely associated with HCC, however, the risk of malignant transformation from FNH to HCC has been already suggested[9]. In this study, we describe a 74-year-old female patient with an FNH and a concomitant HCC nodule in a non-cirrhotic liver. The nodule consisted of two HCC components with distinct histopathological features. We demonstrated, using whole-exome sequencing (WES), that the FNH and the HCC share a common phylogenetic origin.

## Methods

**Patient**. Liver biopsies and resected material were obtained from the University Hospital Basel. Written informed consent was obtained from the patient for the publication of the case details. The study was approved by the ethics committee of the north-western part of Switzerland (Ethics Committee of Basel, EKBB, numbers 2019-00816 and 2014-099)[10,11].

**Immunophenotypic characterization**. Sequential 3 μm-thick sections of formalin-fixed, paraffin-embedded (FFPE) tumoral tissue were used. Deparaffinized serial sections were stained by Hematoxylin and Eosin (H&E) and Novotny reticulin stain. Histopathologic HCC grading was performed according to the Edmondson system. Immunohistochemical staining were performed with monoclonal antibodies against glutamine synthetase (GS) (clone GS-6, mouse, Ventana/Roche, Mannheim, Germany), Glypican-3 (clone 1G12, mouse, Ventana/Roche, Mannheim, Germany), CD34 (Ventana/Roche, Mannheim, Germany), Ki67 (clone Mib1, catalog no IR626, Dako, Carpinteria, CA, USA), Serum Amyloid-A (SAA) (clone Mc1, catalog no IR605, Dako, Carpinteria, CA, USA) and C Reactive Protein (CRP) (clone ab32412, Abcam, Cambridge, MA, USA) on a Benchmark immunostainer (Ventana, Roche) according to the manufacturers' instructions.

**DNA extraction and whole-exome sequencing**. FFPE tissue of the resected tumor was micro-dissected to separate the FNH and the two HCC components as previously described[12]. DNA was extracted using the Qiagen DNeasy Blood & Tissue kit (Qiagen, Hilden, Germany) according to the manufacturer's instructions.

Extracted DNA from each micro-dissected component and the adjacent non-tumoral liver tissue were separately subjected to WES. The Twist Human Core Exome kit was used for whole-exome capture according to the manufacturer's guidelines. Sequencing was performed on Illumina NovaSeq 6000 using paired-end 100 bp (mean sequencing depth 247× for the FNH, 297× for HCC1, 211× for the HCC2/HGDN, and 231× for germline (adjacent non-tumoral liver tissue). Sequencing was performed by CeGaT (Tübingen, Germany).

Reads obtained were aligned to the reference human genome GRCh38 using Burrows-Wheeler Aligner (BWA, v0.7.12)[13]. Local realignment, duplicate removal, and base quality adjustment were performed using the Genome Analysis Toolkit (GATK, v4.1, and Picard (http://broadinstitute.github.io/picard/). Somatic single nucleotide variants (SNVs) and small insertions and deletions (indels) were detected using Mutect2 (GATK 4.1.4.1)[14] and Strelka v.2.9.10[15]. Only variants detected by both callers were kept. SNVs and indels outside of the target regions (i.e., exons), those with a variant allelic fraction (VAF) of <5% and/or those supported by <3 reads were filtered out. Variants for which the tumor VAF was <5 times that of the paired non-tumor VAF were excluded, as were variants identified in at least two of a panel of 123 non-tumor samples, captured and sequenced using the same protocols using the artifact detection mode of MuTect2 implemented in GATK. All indels were manually inspected using the Integrative Genomics Viewer[16]. To account for the presence of somatic mutations that may be present below the limit of sensitivity of somatic mutation callers, we used GATK Unified Genotyper to interrogate the positions of all unique mutations in all samples to define the presence of additional mutations.

FACETS v.0.5.14[17] was used to identify allele-specific copy number alterations. Genes with total copy number greater than gene-level median ploidy were considered gains; greater than ploidy +4, amplifications; less than ploidy, losses; and total copy number of 0, homozygous deletions. Somatic mutations associated with the loss of the wild-type allele (i.e., loss of heterozygosity [LOH]) were identified as those where the lesser (minor) copy number state at the locus was 0. The cancer cell fraction (CCF) of each mutation was identified using ABSOLUTE v. 1.0.6[18].

**Mutational signatures**. Decomposition of mutational signatures was performed using deconstructSigs[19] based on the set of 60 mutational signatures ("signatures.exome.cosmic.v3.may2019")[20,21].

**PCR amplification, Sanger sequencing, and quantitative real-time PCR**. For the identification of hotspot somatic mutations in TERT promoter, primer sets that amplify the hotspot sites of the TERT promoter were designed as previously described[22] and are available in our previously published study[23]. PCR amplification was performed from 100 ng of genomic DNA using the AmpliTaq Gold 360 Master Mix Kit (Life Technologies) on a SimpliAmp Thermal Cycler (ThermoFisher) as previously described[23]. Sequencing was performed using purified PCR fragments (QIAquick PCR Purification Kit, Qiagen) on an ABI 3730 capillary sequencer using the ABI BigDye Terminator chemistry (v3.1, Life Technologies). Sequences of the forward and reverse strands were analyzed using 4Peaks (https://nucleobytes.com/4peaks/). All analyses were performed in triplicate.

RNA extraction from FFPE tissues was performed using RecoverAll Total Nucleic Acid Kit for FFPE (ThermoFisher) according to manufacturer's guidelines. Quantitative RT-PCR analysis was performed using SYBR Green. GAPDH was used as housekeeping genes for normalization. mRNA fold expression change was calculated by the 2-ΔΔCT method as previously

described[24]. The following Primers set were used: *GAPDH* Foward 5′ -AGGTGAAGGTCGGAGTCAACG-3′ and Reverse 5′ -TGGAAGATGGTGATGGGATTT-3′ and *TERT*[25] Foward 5′ -GCCGATTGTGAACATGGACTACG-3′ Reverse 5′ -GCTCGT AGT TGAGCACGCTGAA-3′.

**Clonality analysis**. Clonal prevalence analysis was conducted using the hierarchical Bayesian model PyClone, which estimates the cellular prevalence of mutations in deeply sequenced samples, using allelic counts, and infers clonal structure by clustering these mutations into groups with co-varying cellular frequency. PyClone was run using a two-pass approach, whereby mutations whose cellular prevalence estimate had standard deviation >0.3 were removed before a second pass analysis was run. A cellular prevalence of >80% was used as a threshold for clonality.

## Results

**Patient information and clinical history**. A 74-year-old female patient with a history of continued alcohol abuse (one bottle of wine/day for more than 10 years; 9 units/day) and malnutrition (BMI 12.8 kg/m$^2$) presented in the emergency department with multiple fractures of the femur and pelvis after falling at home. Initial computed tomography (CT) incidentally revealed an $18 \times 17$ mm hypervascular nodule in liver segment 8 in November 2019 (Fig. 1a), the lesion showed complete wash-out in the venous/delayed phase and was therefore classified as LI-RADS 4 (probably HCC, biopsy recommended)[26]. Standard liver function screening revealed normal aminotransferases (ASAT 29 U/l, normal range 11–34 U/l; ALAT 18 U/l, normal range 8–41 U/l), bilirubine (9.2 µmol/l, normal range < 15 µmol/l), INR 0.9, normal range < 1.3) and albumin levels (38 g/l, normal range 35–52 g/l). Magnetic resonance imaging for better characterization of the nodule was declined by the patient. Therefore, an ultrasound-guided needle biopsy was performed and a detailed histological assessment, including morphologic and immunohistochemical analysis, was consistent with the diagnosis of FNH without signs of cirrhosis or malignancy in the background liver (Fig. 2a). A follow-up thoracoabdominal CT scan after ~7 months (June 2020) showed an increase in nodular size up to $21 \times 32$ mm, with inhomogeneous arterial phase hyperenhancement and venous/delayed phase washout (Fig. 1b). There was a nodule-in-nodule pattern and a threshold growth of more than 50% within 6 months, both features supporting an upgrade to LI-RADS 5 (definitive HCC). Alpha-fetoprotein was not elevated at 4.7 kIU/l (normal range < 5.8 kIU/l) and showed no relevant increase over time. A new pulmonary focus was demarcated in the right upper lobe in November 2019, which was, however, assessed as indeterminate.

Due to the imaging findings supporting a definitive HCC diagnosis, the patient was scheduled for surgical resection in accordance with the Barcelona clinic liver cancer staging[27] management guidelines (stage A—early stage). Preoperative risk stratification showed neither portal hypertension (portal vein pressure gradient 3 mmHg) nor esophageal varices and complete resection of the tumor was performed without any complications. The patient now undergoes regular CT scan controls. The last CT scan in September 2021 did not show any metastatic suspicion.

**Pathologic characterization of the lesions**. The first macroscopical analysis of the resected specimen revealed a well-circumscribed, but not encapsulated lobulated solid mass of 2.9 cm in diameter consisting of two different components. The bigger component was yellow with a fibrous scar and focal hemorrhage while the other part was white and firm (Fig. 2b).

Microscopic examination of the specimen revealed the presence of two different lesions classified as an HCC (Edmondson grade 2; HCC1) and an early HCC/High-grade dysplastic nodule (Edmondson grade 1 HCC or High-grade dysplastic nodule; HCC2/HGDN) adjoining the FNH nodule (Fig. 2b). The FNH nodule showed the classical histologic picture with numerous foci of hepatocytes intersected with arteria and bile ductuli-rich fibrous bands (Fig. 2c and Supplementary Fig. 1l). The reticulin staining demonstrated a preserved reticulin framework inside the nodule. Immunohistochemical stains for GS showed the distinctive "map-like" patchy staining of the hepatocytes. Glypican-3 (GPC3) was negative. The nodule was negative for SAA while partial positivity observed for the CRP (Supplementary Fig. 1d-e). CK19 highlighted the presence of numerous bile ducts within fibrotic bands of the nodule (Supplementary Fig. 1k). The proliferation index measured by Ki67 was lower than 1%. (Supplementary Fig. 1g). The two neighboring lesion components had distinct morphological features from each other. The larger tumor, 15 mm in diameter, showed Edmondson grade 2 trabecular-solid pattern HCC with steatotic cellular change (HCC1; Fig. 2d). GS and GPC3 immunostainings displayed diffuse positivity and the reticulin framework was distinctly lost. The adjoining smaller tumor, 8 mm in diameter, vaguely nodular, consisted of monotonous hepatocytes of nearly normal appearance with focal atypia (Fig. 2e). Reticulin staining revealed focal thickened trabeculae and loss of reticulin framework. GS was patchy positive while GPC3 was expressed by small subsets of tumoral cells. In HCC components, CD34 staining revealed capillarization of sinusoids (Supplementary Fig. 1a-b). The Ki67 proliferative index was less than 1% for the tumoral cells on both of the components (Supplementary Fig. 1h-i). By microscopical analysis, the differentiation of early-stage well-differentiated (Edmondson grade 1) HCC and high-grade dysplastic nodule (HCC2/HGDN; Fig. 2e). The background liver was free of pathology (Supplementary Fig. 1c). The postoperative tumor stage was pT1a pN0 G1 R0.

The patient was included in a clinical study, investigating pembrolizumab vs. placebo in an adjuvant setting for patients with a high risk of HCC recurrence.

**Genomic characterization reveals clonal evolution of the FNH to HCC**. To better understand the origin of these lesions and to investigate the possibility of a clonal relationship between the FNH and the two HCC components, we performed high-depth WES of the separately microdissected lesions (FNH, HCC1: Edmondson grade II HCC and HCC2/HGDN: Edmondson grade I HCC/ High-grade dysplastic nodules), along with non-tumoral tissue used to call somatic genetic alterations excluding germline variants. We detected 94, 102, and 101 non-synonymous mutations in the FNH sample, HCC1, and HCC2/HGDN, respectively. Of these, 80 were common between the FNH and both HCCs, and 6 were shared between the HCCs and absent in the FNH (Fig. 3a and Supplementary Data 1). Ten, 16, and 11 mutations were found exclusively in the FNH, HCC1, or HCC2/HGDN, respectively (Fig. 3a). Four mutations were found in the FNH and HCC2/HGDN, while no mutations were shared between only HCC1 and the FNH exclusively. In addition, given that the WES does not cover the promoter region of *TERT*, we performed Sanger sequencing for the two hotspot mutations commonly found in HCC ( $-c.124 \, C > T$ and $-c.146 \, C > T$). We found that the FNH and the two HCC components harbor the hotspot mutation $-c.124 \, C > T$. The effect of this mutation was also investigated at the *TERT* mRNA level. *TERT* expression was higher in the lesions compared to the matched non-tumoral liver

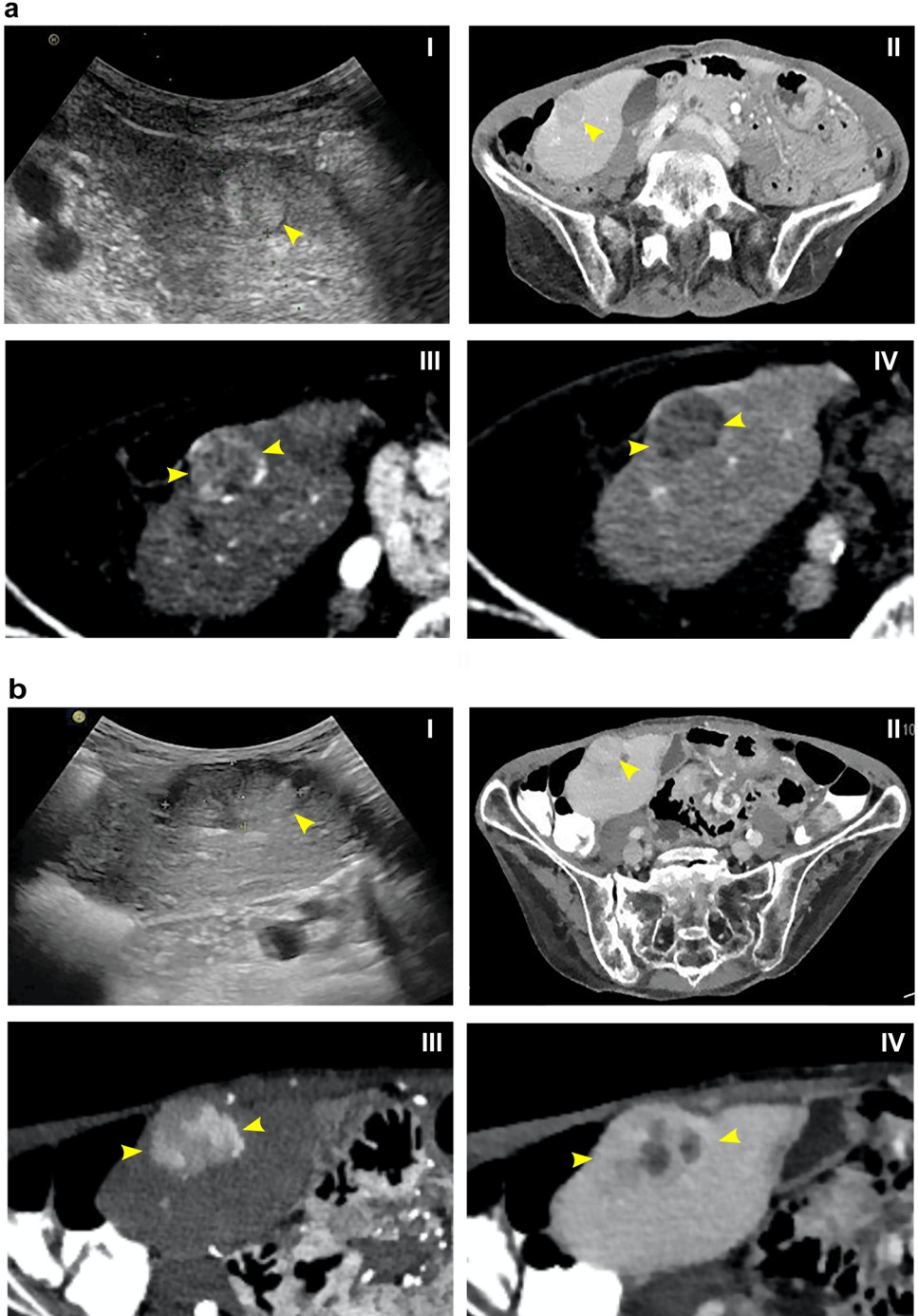

**Fig. 1 Radiological imaging of the liver nodule at different time points. a** In the initial imaging, abdominal ultrasound shows a solitary, roundish nodule with a mosaic pattern, in the liver (I). Axial contrast computerized tomography (CT) scan (II) of the abdomen shows a partially cystic nodule (maximum diameter 18 mm) in liver segment V. The arterial phase shows a hyperenhancing lesion (III), the portovenous phase demonstrates typical wash-out in comparison to surrounding liver tissue (Liver Reporting & Data System (LI-RADS) 4 classification) (IV). **b** A follow-up ultrasound (I) and contrast CT scan (II) were performed. The arterial phase CT scan shows a 32 mm hyperenhancing lesion with inhomogeneous contrast up-take, threshold growth more than 50%. The portovenous phase demonstrates wash-out and nodule-in-nodule pattern (LI-RADS 5 classification). Arrows highlight the tumor nodules.

(Supplementary Fig. 2). The mutations shared by all samples included missense variants in *ATG5* (T214A) coupled with the LOH of the other allele. Of note, alterations in this gene were previously reported in the pathogenesis of FNH in mouse models[28]. Other shared mutations were observed in *ANGPT1* (Q162K), the expression of which has been found to be dysregulated in other FNH lesions[8], and in *HNRNPA2B1* (D164G), which has been reported as a hepatic mutational cancer driver[29].

In addition, we detected somatic mutations in well-known oncogenes such as *CBL* (R822W) and *G6CP* (A161S) that have previously been described in HCC and other cancer types[29,30] (Fig. 3b, Supplementary Data 1). On the other hand, the mutations unique to the HCC samples included somatic alterations in *bona fide* cancer driver genes such as *MYCN* (G46V) and *MAP2K4* (Q118K)[29] (Fig. 3b). In addition, given that the clinical history might suggest alcohol-related liver injury we analyzed the

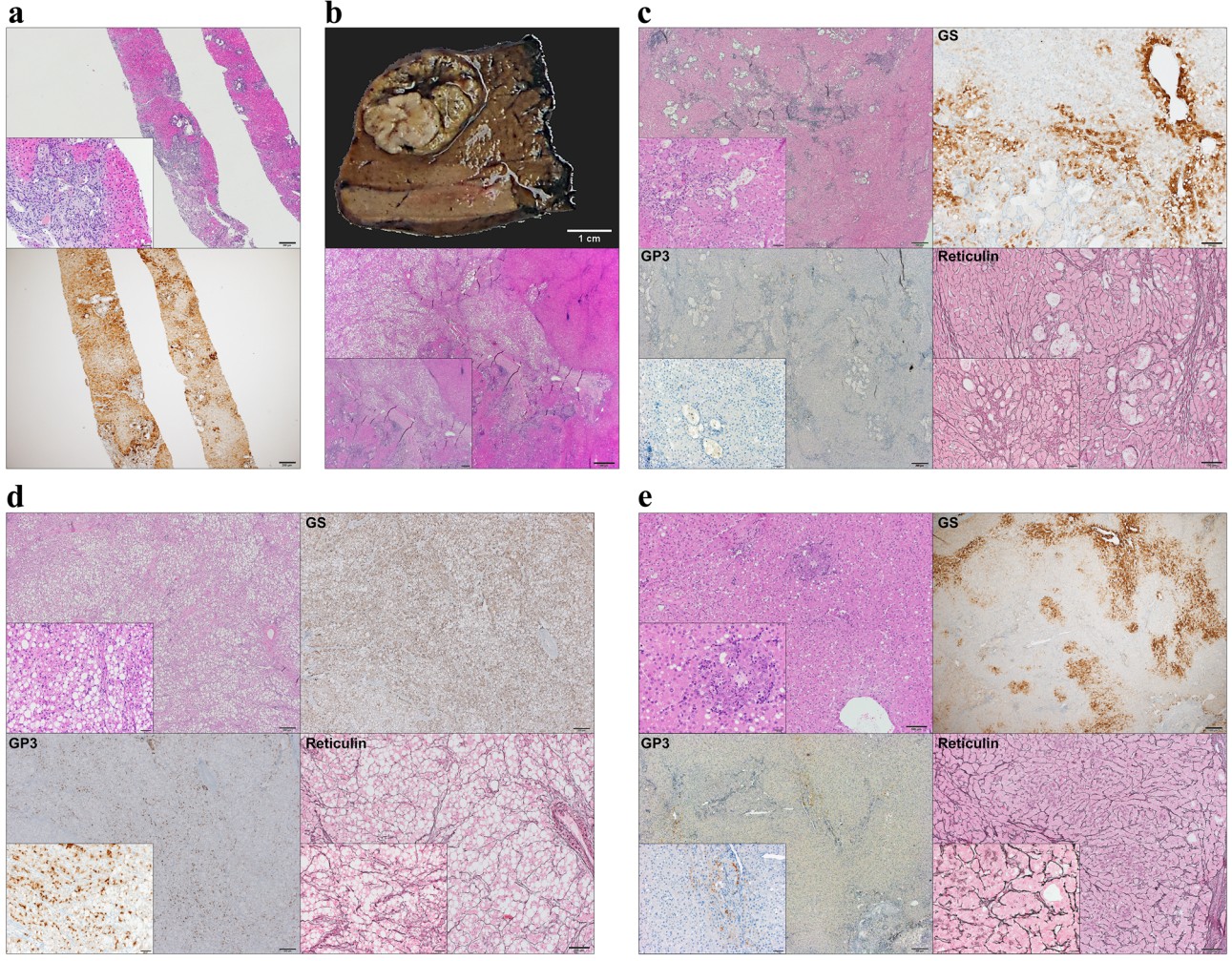

**Fig. 2 Histopathological characterization of the Focal Nodular Hyperplasia (FNH) and Hepatocellular Carcinoma (HCC). a** Hematoxylin and eosin (H&E) staining in the diagnostic biopsy (liver segment 6) shows hyperplastic tissue with dilated bile duct proliferation (scale bar 200 μm; insert 50 μm). Glutamine synthetase (GS) staining positive was a "map-like" pattern (scale bar 200 μm). **b** The nodule macroscopically consists of two different components (scale bar 1 cm). H&E staining of the nodule showed two different HCC components adjoining the FNH nodule (scale bar 500 μm; insert 200 μm). **c** Histological analysis of FNH showing classical features of FNH, such as nodular formation and fibrous septa with abnormal vessels (scale bar 200 μm; insert 50 μm). GS staining displayed the typical "map-like" pattern consistent with the diagnostic biopsy (scale bar 100 μm; insert 50 μm). Glypican 3 (GPC3) staining was negative (scale bar 200 μm; insert 50 μm), the reticulin framework was preserved (scale bar 100 μm; insert 50 μm). **d** Histological analysis of HCC1 (Edmondson grade II). Steatohepatitic variant HCC with the presence of large-droplet steatosis, with fibrosis and Mallory-Denk bodies (scale bar 200 μm; insert 50 μm). GS staining showed diffuse positivity (scale bar 200 μm; insert 50 μm) while GPC3 was positive in the majority of the malignant cells (scale bar 200 μm; insert 50 μm). The tumor showed definitive loss of reticulin (scale bar 100 μm; insert 50 μm). **e** Histological analysis of HCC2/HGDN (Edmondson grade I HCC or High-grade dysplastic nodule; HCC2/HGDN). HCC consists of monotonous cell proliferation in a trabecular pattern (scale bar 100 μm; insert 20 μm). GS staining displayed patchy positivity (scale bar 200 μm;) as well as GPC3 whose expression was detected in focal areas (scale bar 200 μm; insert 50 μm). Partial loss of the reticulin framework was found in the tumor (scale bar 100 μm; insert 20 μm).

presence of the COSMIC single base substitution signature 16 (SBS16) that has been correlated with alcohol consumption[30,31]. This analysis did not find evidence for this signature in the repertoire of synonymous and non-synonymous mutations detected in our case (0% FNH, 0.14% HCC1, 0.13% HCC2/HGDN). Copy number analysis reflected the analysis of coding mutations; the majority of alterations were shared between all components, while HCC1 acquired further alterations such as loss of 1p, and gain of 8q including the *MYC* locus (Fig. 3c).

Of note, the genomic analysis revealed the peculiarity of these lesions. We detected no common driver genetic somatic alterations usually found in HCC or inflammatory hepatocellular adenomas (IHA), such as mutations in *CTNNB1*, *TP53*, *AXIN1*, or *ARID1A*, *IL6ST*, *GNAS*, *STAT3* neither in the FNH nor in either HCC components. This suggests the HCC associated with the FNH, in this case, may have emerged through a unique tumorigenic process.

We then performed a clonality analysis using all synonymous and non-synonymous mutations to determine how the FNH is clonally related to the *bona fide* HCC components. We found that all lesions consisted of multiple cell populations (Fig. 3d). A clonal cell population containing a cluster of mutations was present in all three components, which included alterations affecting *CBL, ANGPT1,* and *ATG5* (Fig. 3d). Clones harboring a cluster of mutations including *MYCN* and *MAP2K4* mutations were absent from the FNH, and clonal in HCC1, and just below the threshold in HCC2/HGDN (CCF 0.796) (Fig. 3d). Interestingly, *MAP2K4* has been identified as a mutational cancer driver

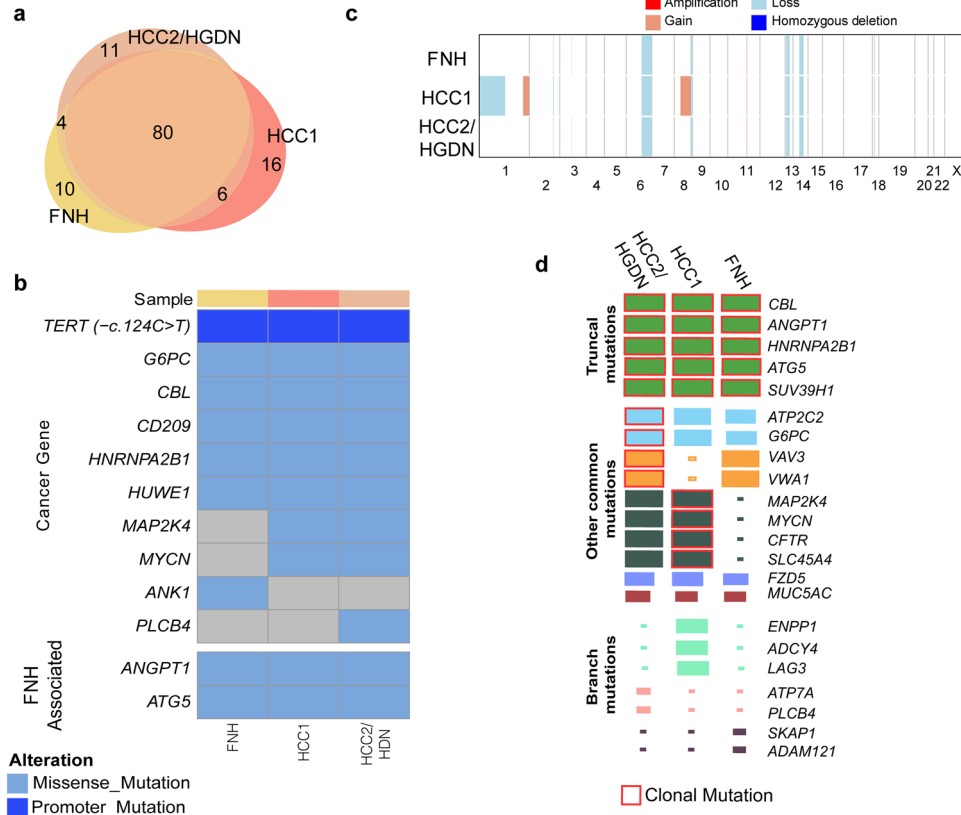

**Fig. 3 Genetic characterization of Focal Nodular Hyperplasia (FNH) and two associated Hepatocellular Carcinoma (HCC) components. a** Venn diagram representing the number of somatic mutations detected in each lesion subjected to whole-exome sequencing (WES) (**b**) Oncoprint of genetic alterations detected in FNH, and associated HCC components by WES. Alterations are colored according to the legend. Alterations shown are those included in cancer gene lists (see Supplementary Data 1). **c** Summary of genome-wide copy number alterations detected by WES. Copy number changes are colored according to the legend. **d** Chart showing clonality of selected mutations in each cluster. Size of square denotes cancer cell fraction of cluster.

in cholangiocarcinoma[29,32]. The FHN and HCC2/HGDN contained clones with a cluster of mutations including the *VAV3* and *VWA1* genes which were not present in HCC1, but present at a CCF of 0.72 and 0.71 in the FNH and HCC2/HGDN, respectively. In HCC1, however, cells containing the *ADCY4*, *ENPP1*, and *PDZD7* mutations expanded to a CCF of 0.69. These data demonstrate the clonal relatedness of the FNH and HCC components, and the divergent evolution of the FNH and the two HCC components.

## Discussion

FNH is considered to be a benign process, resulting from localized hyperperfusion of the parenchyma due to arterial malformations, which subsequently induces hepatocellular hyperplasia[2]. It is thought that FNH has no malignant potential and is rarely associated with synchronous HCC. Nonetheless, a few articles in the literature reported patients with synchronous FNH and HCC without data showing their genetic relationship[4–7]. Here we performed a genetic analysis of one FNH with two associated lesions classified as Edmondson grade I HCC or high-grade dysplasia and an Edmondson grade II HCC components and found that the FNH is composed of multiple clones at diagnosis. We think our results suggest the FNH likely progressed to HCC through clonal selection and/or the acquisition of additional genetic events.

The histological view of FNH nodule consists of numerous foci of hepatocytes intersected with arteria and bile ductuli-rich fibrous bands, which is diagnostic for FNH together with the distinctive "map-like" patchy GS expression of the hepatocytes. Given the possible morphological similarity of FNH with inflammatory type hepatocellular adenoma (IHA) in some cases, we performed additional immunostains (SAA and CPR) to rule out this alternative diagnosis. The immunostaining revealed negativity for SAA and partial positivity for CPR. A study performed by Joseph et al. has shown that the SAA expression is positive in the vast majority of IHA (92.6%) while the CRP expression was found in 78% of FNH[33]. The diagnosis of IHA was then excluded based on these results together with other morphological features. These observations were further supported at the genomic level, given the absence of common driver genetic somatic alterations usually found IHA such as mutations in *CTNNB1*, *IL6ST*, *GNAS*, and *STAT3*[34].

Peritumoral hyperplasia (PTH) is another entity that can resemble FNH. Arnason et al. described PTH as a hyperplastic response to increased blood flow in the peritumoral parenchyma. Its characteristic morphology is a rim of hepatocytes surrounding the circumference of HCC like a cuff[35]. In our case, the lesion was a nodular lesion localized at the neighbor of the tumor, instead of encircling the HCC. Thus, PTH was not considered for the diagnosis.

Given its rarity, there is a limited effort to investigate the genomic features of FNH. Most studies have, so far, focused on clonal analysis using the HUMARA test and showed polyclonality of FNH which supports its reactive hyperplastic rather than neoplastic nature[5,6,36,37]. Moreover, three comparative genomic hybridization studies showed that 14–50% FNHs displayed frequent gains and losses at specific chromosome arms

such as 1q and 22q[7,38,39]. Interestingly, Chen et al. failed to detect common HCC mutations on 11 FNHs with genomic sequencing of *TP53*, *CTNNB1*, *AXIN1*, and *APC* which is not that astonishing considering the fact that—as exemplified by our case—the molecular mechanism of FNH hepatocarcinogenesis might be different from the mechanism/s observed in HCCs not associated with FNH[40]. Furthermore, two quantitative gene expression studies showed alteration of angiopoietin expression levels[8,41], supporting the importance of vascular alterations in the pathogenesis of FNH[1].

Autophagy and endocytosis are cellular pathways responsible for the degradation and recycling of intracellular and extracellular components, respectively, with a strong role in tumor promotion[42,43]. We found that *ATG5*, a critical regulator of autophagy, and *CBL*, a regulator of receptor tyrosine kinases by endocytosis, were among the genes found mutated in all samples (FNH and 2 HCC components). Experiments in mice with systemic mosaic deletion of *Atg5*[44] or liver-specific *Atg5* knock-out[28] reported the development of spontaneous benign tumors in the liver such as FNH as a result of the impairment of autophagy that led to oxidative DNA damage and hepatocyte proliferation. The full development of HCC in *Atg5*-deficient livers required a more permissive genome with the suppressed expression of tumor suppressors[28]. Indeed, a recent study published by Barthet et al.[45] showed that loss of *ATG5* in the context of hemizygosity of *PTEN* in mice causes the development of HCC involving ductular reaction[45]. Mechanistically, the authors showed that loss of autophagy is followed by activation of YAP/TAZ in hepatocytes leading to its differentiation into biliary-like liver progenitor cells (ductular reaction) that ultimately lead to HCC.

Interestingly, similarly to our result, a previous study including 7 FNH cases with 1 synchronous HCC reported the loss at 6q11.1-q23 in one of the FNH cases that includes the locus of *ATG5*[7]. In our study, we found that the *ATG5* mutation (coupled with LOH of the other allele) was accompanied by the presence of clonal mutations in *MYCN* and *MAP2K4* in both HCC samples but these were absent from the FNH. While *MYCN* is a well-recognized oncogene in several cancers including HCC[46], *MAP2K4* is reported to act as a tumor suppressor in HCC[47]. The absence of a common HCC driver alteration affecting *CTNNB1*, *TP53*, or *ARID1A*[47], together with alterations in *ATG5* and *CBL* and in *MYCN* and *MAP2K4* may point to a unique path of FNH-associated HCC development.

While several studies have reported co-occurrence of FNH and HCC[4–6,9], to our knowledge this is the first study showing a clonal relationship between these lesions. Moreover, we found uncommon genetic alterations in the HCC that might be associated with hepatocarcinogenesis in the background underlying FNH. Importantly, our results suggest in extremely rare cases, FNH can share some similar genomic alteration with HCC, may not be absolutely benign and may, albeit rarely, progress to HCC. Although our results suggest that FNH was a non-obligate precursor lesion of HCC, another hypothesis is that given that HGDN and HCC are arterialized lesions, the disrupted local vascular flow may lead to the development of FNH. We cannot fully exclude the possibility that the FNH developed in an area of an HCC precursor that already harbored the genetic alterations. However, given the clonal frequencies of the observed mutations, we think it is more likely that the FNH progressed to HCC through clonal selection.

Current clinical guidelines[48] recommend a conservative approach without any follow-up, treatment should be pursued only in exceptional cases[48]. Our results may suggest that further studies may help pinpoint features of FNHs that indicate potential for progression thus helping to refine the surveillance strategy for these lesions.

**Reporting summary**. Further information on research design is available in the Nature Research Reporting Summary linked to this article.

## Data availability

Sequencing data are available on the European Genome-Phenome Archive database under the accession number EGAD00001007702.

Digital pathology images have been deposited on the Zenodo database under the accession number https://doi.org/10.5281/zenodo.555433[49].

Source data for graph in Supplementary Fig. 2b are available in the Supplementary Data 2 file.

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

## Acknowledgements

C.K.Y.N., and S.P. were supported by the Swiss Cancer League (KFS-4543-08-2018, KFS-4988-02-2020-R, respectively); L.M.T., was supported by AIRC grant number IG 2019 Id.23615. S.P. was supported by the University of Basel (Research Fund Junior Researchers and Department), by the Krebsliga Beider Basel (KLbB-4473-03-2018), by the Theron Foundation, Vaduz (LI), by the Surgery Department of the University Hospital Basel and by the The Prof. Dr. Max Cloëtta foundation. The funders had no role in study design, data collection, and analysis, decision to publish, or preparation of the paper.

## Author contributions

S.P. and O.K. conceived the study. S.P. supervised the study. J.G and C.K.Y.N. performed the bioinformatic analysis; C.E., J.V., A.T., M.S.M., L.D.T. and L.M.T. performed the histopathologic evaluation; L.F., G.F.H., S.T-M., T.B. M-A.M., M.H.K.H., M.vF., M.H.H., S.D.S., and O.K. performed the clinical evaluation and surgical procedure on the patient. C.E., M.C.-L., and M.M. processed the sample for the WES; C.E., M.C-L, J.G., and L.F. discussed the data and wrote the first draft of the paper that was revised by C.K.Y.N. and S.P. All authors have read and revised the paper.

## Competing interests

M.S.M. has received speaker's honoraria from Thermo Fisher and honoraria as an advisory board member from Novartis. The other authors declare no competing interests.
