## [Peer Review File · Communications Medicine]

Reviewers' comments:

Reviewer #1 (Remarks to the Author):

The case report entitled 'Genomic Analysis of Focal Nodular Hyperplasia with Associated Hepatocellular Carcinoma Unveils its Malignant Potential' by Ercan et al. describes an interesting and clinically provocative case. Using whole exome sequencing they describe a multiloculated liver lesion containing elements of FNH and HCC which phylogenetically are related. This implies that FNH may have direct malignant potential. Intriguingly this appears to have given rise to two HCC-like lesions; possibly independently.

Major comments

The images presented in Fig 2e are not convincing themselves that that lesion is HCC rather than a further FNH. Can additional pathology images be provided in a revised manuscript and ideally deposited publically alongside sequencing data at the time of publication? A proliferation marker would be interesting to include in Fig2.

Clearly the gathering of additional cases with similar findings is beyond the scope of this report but it should be highlighted that on the basis of this single case there is a debatable evidence to change clinical practice. The authors propose a clinical implications of this work is engaging patients with FNH in "monitoring", this clinical recommendation should be more carefully nuanced. Are the authors suggesting that a non cirrhotic with an FNH should be engaged in long term cross sectional imaging based surveillance potentially for many years? This appears to be the conclusion but I believe that whilst this is an interesting report there is insufficient data to assess the relative risk of HCC in this population versus a cirrhotic at risk patients and long term FNH follow up is likely to require repeated CT (with associated risks) or MRI (intolerable to this index patient). Therefore, I would suggest a more clinically cautious conclusion to this report. I would suggest referencing major international guidelines e.g. AASLD or EASL on the management of such lesions to guide the reader to accepted practise for monitoring (or not) of FNH.

This case represents a single well characterised and clinically annotated case. The clinical history might suggest alcohol related liver injury. Can the alcohol consumption be quantified both in consumption and years of consumption? Was there a mutational signature that would correspond to this even in the absence of histopathological evidence of advanced fibrotic chronic liver disease?

Minor comments

Line 67 Highlight imaging with ultrasound and CT was performed in figure 1.

Please outline in more detail the radiological components of the Liver lesion justifying LiRADs 4

The figures could be made clearer to follow. Please state what the arrows show in Figure 1a. Also suggest highlight which phase the scans and the orientation of the sections would be helpful for the general reader (coronal and sagittal if using different orientations); however I would also suggesting showing the arterial and portal phase in the same orientation in Fig 1a for direct comparison of the phases. Can the interval of within 6 months be more specific? Please highlight that GS is the lower panel in Fig 2a. Highlight the liver segment number might be helpful from a surgical perspective. If the same colours are to match between 3a and b should be a clearer match in tone, currently it is unclear what the colours add to the labels beneath in 3b.

It is assumed that CT based chest staging was also performed. Can this be reported also please as evidence of standard clinical practice for exclusion of radiologically apparent metastasis?

The statement regarding male patients is irrelevant in this case (line 247) and can be removed.

With reference to a mechanistic role of autophagy and HCC, Barthelet et al. (DOI:

10.1126/sciadv.abf9141) recently demonstrated a role for ATG5/7 loss in promoting HCC in a

steatotic murine model of HCC. I would suggest also discussion related to this study in the associated paragraph discussed autophagy mediated carcinogenesis (line 175).

Minor Typos

Line 51 "that present in HCC"

Line 68 "aminotransferase(s)"

The sentence commencing line 123 should be broken up to make it more readable.

Reviewer #2 (Remarks to the Author):

This manuscript reported a 74-year-old female patient with an FNH and associated concomitant hepatocellular carcinoma. Immunophenotypic and exome sequencing showed that HCC may developed from FNH.

Major questions:

- 1) High grade dysplastic nodule and early HCC is very similar. Which evidences show that HCC2 is not dysplastic nodule ?
- 2) TERT alteration (promoter mutaiton, copy number variation, HBV insertion) is usually occured in early progress of HCC. Whether authors observe TERT alteration? If whole exome sequencing can not cover the promoter region of TERT, expression of TERT should be tested by IHC or other approaches. Whether TERT has higher expression in HCC1 and HCC2 than in FNH ?

Minor questions:

- 1) The raw data of exome sequencing should be submitted to a public database (such as NCBI SRA).

Reviewer #3 (Remarks to the Author):

The authors present a case of concurrent hepatocellular carcinoma (HCC) and focal nodular hyperplasia (FNH) arising in non-cirrhotic liver. FNH is typically considered to be a non-neoplastic lesion formed in reaction to local vascular flow aberrations. The authors show a lesion which has 3 distinct histomorphologies, two of which are definitely HCC. They have evaluted the areas with a small panel of immunostains (glutamine synthetase, CD34, glypican-3), as well as reticulin. They went on to microdissect the separate areas and perform whole exome sequencing. They identified 80 mutations shared between the 3 areas. Clonality assay demonstrated multiple clones in each area. The authors conclude that their findings support the hypothesis that FNH can be a precursor to HCC.

1. My major concern with the paper is in regard to the "FNH". Specifically, I am concerned that this may actually represent an inflammatory hepatocellular adenoma (which would be a known precursor lesion for HCC and can result in HCC in non-cirrhotic liver). These can very much resemble FNH, with presence of ductules, inflammation, and pseudoportal tracts (even sometimes septa); in fact, before they were discovered to be adenomas, they were called "telangiectatic FNH". Based on the photomicrographs provided and the glutamine synthetase stains shown (I am not convinced of a

map-like/geographic pattern -- and HCAs can have glutamine synthetase staining, including perivenular), I think more clarification is required. Immunostains for CRP and SAA should be performed. Mutational analysis, if not already performed, for mutations in IL6ST, GNAS, STAT3, and CTNNB1 should be completed (if already completed, the findings should be specifically described).

2. The adjacent non-lesional liver should also be addressed. It would be of particular interest to perform similar mutational analysis on the non-lesional liver right next to the tumour, and then more remotely. If similar mutations are seen in the background liver and the FNH-like area, then the support this as a precursor lesion is lessened; however, if no mutations are found in the background, then the conclusion (assuming I-HCA has been ruled out) is supported.

3. The authors should consider review and discussion of peritumoural hyperplasia, which is a FNH-like lesion. If they feel their FNH-like area is not peritumoural hyperplasia (and, if proven not to be inflammatory hepatocellular adenoma), they should make a statement defending their decision.

The authors may find this article helpful: Arnason T, Fleming KE, Wanless IR. Peritumoral hyperplasia of the liver: a response to portal vein invasion by hypervascular neoplasms. *Histopathology*. 2013;62:458-64.

Overall, I think more work needs to be performed to support the conclusion and strengthen the paper. If the authors can further support that the FNH-like area truly is FNH and that the background liver harbours no significant mutations, then this may be paradigm changing (a lesion thought to be non-neoplastic and not a precursor may indeed be!) and affect patient care (perhaps more resections, more radiographic follow-up).

We are delighted that the Reviewers and the Editorial Board of Communication Medicine found our manuscript meritorious. We would like to thank all reviewers for their constructive comments and suggestions that have helped improve our manuscript and given us a valuable opportunity to clarify our interpretation of the results.

Below you can find a point-by-point response to the reviewers' comments:

Reviewers' comments:

Reviewer #1 (Remarks to the Author):

The case report entitled 'Genomic Analysis of Focal Nodular Hyperplasia with Associated Hepatocellular Carcinoma Unveils its Malignant Potential' by Ercan et al. describes an interesting and clinically provocative case. Using whole exome sequencing they describe a multiloculated liver lesion containing elements of FNH and HCC which phylogenetically are related. This implies that FNH may have direct malignant potential. Intriguingly this appears to have given rise to two HCC-like lesions; possibly independently.

Authors: We thank the reviewer for the favorable assessment of our work.

Major comments

The images presented in Fig 2e are not convincing themselves that that lesion is HCC rather than a further FNH. Can additional pathology images be provided in a revised manuscript and ideally deposited publically alongside sequencing data at the time of publication? A proliferation marker would be interesting to include in Fig2.

Authors: We thank the reviewer for the observation. In the revised version of the manuscript, we increased the resolution of the previously included micrographs and we replaced some of the previous ones. Moreover, we have performed additional immunostaining as requested by the reviewers (#1 and #2) and added representative micrographs in **Fig. S1**.

Following the suggestion made by this reviewer, we have provided digital slides from two representative blocks for the following stains (links can be used to access the images scan at full resolution for review):

1. H&E-1
2. H&E-2
3. GS-1
4. GS-2
5. Novotny-2
6. Novotny-1
7. SAA-2
8. SAA-1
9. CRP-2
10. CRP-1
11. Ki67-1
12. Ki67-2

These digital images have been also deposited on the Zenodo database under the accession number **10.5281/zenodo.5554337**

The new analysis has been integrated into the revised version of the manuscripts as follow:

Results: the paragraph “Pathologic characterization of the lesions” on pages 5 and 6 has been updated with the new data and with the new nomenclature suggested by reviewer #2

Material and Methods (Immunophenotypic characterization) page 11 paragraph 2: Sequential 3 µm-thick sections of formalin-fixed, paraffin-embedded (FFPE) tumoral tissue were used. Deparaffinized serial sections were stained by Hematoxylin and Eosin (H&E) and Novotny reticulin stain. Histopathologic HCC grading was performed according to the Edmondson system. Immunohistochemical staining were performed with monoclonal antibodies against glutamine synthetase (clone GS-6, mouse, Ventana/Roche, Mannheim, Germany), Glypican-3 (clone 1G12, mouse, Ventana/Roche, Mannheim, Germany), CD34 (Ventana/Roche, Mannheim, Germany), Ki67 (clone Mib1, catalog no IR626, Dako, Carpinteria, CA, USA), Serum Amyloid-A (clone Mc1, catalog no IR605, Dako, Carpinteria, CA, USA) and C Reactive Protein (clone ab32412, Abcam, Cambridge, MA, USA) on a Benchmark immunostainer (Ventana, Roche) according to the manufacturers' instructions.

Figures updated/ added: Fig. 2 and Fig. S1

Data availability

Sequencing data are available on the European Genome-Phenome Archive database under the accession number EGAD00001007702.

Digital pathology images have been deposited on the Zenodo database under the accession number <https://doi.org/10.5281/zenodo.5554337>

Clearly the gathering of additional cases with similar findings is beyond the scope of this report but it should be highlighted that on the basis of this single case there is a debatable evidence to change clinical practice. The authors propose a clinical implications of this work is engaging patients with FNH in “monitoring”, this clinical recommendation should be more carefully nuanced. Are the authors suggesting that a non cirrhotic with an FNH should be engaged in long term cross sectional imaging based surveillance potentially for many years? This appears to be the conclusion but I believe that whilst this is an interesting report there is insufficient data to assess the relative risk of HCC in this population versus a cirrhotic at risk patients and long term FNH follow up is likely to require repeated CT (with associated risks) or MRI (intolerable to this index patient). Therefore, I would suggest a more clinically cautious conclusion to this report. I would suggest referencing major international guidelines e.g. AASLD or EASL on the management of such lesions to guide the reader to accepted practise for monitoring (or not) of FNH.

Authors: We thank the reviewer for the suggestion, in the revised version of the manuscript we have toned down the clinical implication and included in the discussion the limitation of this being only a case report. We have included the following paragraph at the end of the discussion:

Discussion page 11 paragraph 1

Current clinical guidelines¹ recommend a conservative approach without any follow-up, treatment should be pursued only in exceptional cases¹. Our results may suggest that further studies may help pinpoint features of FNHs that indicate potential for progression thus helping to refine the surveillance strategy for these lesions.

This case represents a single well characterised and clinically annotated case. The clinical history might suggest alcohol related liver injury. Can the alcohol consumption be quantified both in consumption and years of consumption? Was there a mutational signature that would correspond to this even in the absence of histopathological evidence of advanced fibrotic chronic liver disease?

Authors: We thank the reviewer for this interesting comment. In the revised version of the manuscript, we added the information required on alcohol consumption. Additionally, we analyzed the presence of the COSMIC single base substitution signature 16 (SBS16) that has been correlated with alcohol consumption^{2,3}. This analysis did not find evidence for this signature in the repertoire of synonymous and non-synonymous mutations detected in our case (0% FNH, 0.14% HCC-1, 0.13% HCC-2). We have included this information in the revised manuscript this figure as follow:

Results page 4 paragraph 1

A 74-year-old female patient with a history of continued alcohol abuse (one bottle of wine/ day for more than 10 years; 9 units/day) and malnutrition (BMI 12.8 kg/m²) presented in the emergency department with multiple fractures of the femur and pelvis after falling at home.

Results page 7 paragraph 1

Additionally, given that the clinical history might suggest alcohol-related liver injury we analyzed the presence of the COSMIC single base substitution signature 16 (SBS16) that has been correlated with alcohol consumption^{2,3}. This analysis did not find evidence for this signature in the repertoire of synonymous and non-synonymous mutations detected in our case (0% FNH, 0.14% HCC1, 0.13% HCC2/HGDN).

Materials and Methods page 13 paragraph 2

Mutational Signatures: Decomposition of mutational signatures was performed using deconstructSigs⁴ based on the set of 60 mutational signatures ("signatures.exome.cosmic.v3.may2019")^{5,6}.

Minor comments

Line 67 Highlight imaging with ultrasound and CT was performed in figure 1.

Please outline in more detail the radiological components of the Liver lesion justifying LiRADs 4

The figures could be made clearer to follow. Please state what the arrows show in Figure 1a. Also suggest highlight which phase the scans and the orientation of the sections would be helpful for the general reader (coronal and sagittal if using different orientations); however I would also suggest showing the arterial and portal phase in the same orientation in Fig 1a for direct comparison of the phases. Can the interval of within 6 months be more specific?

Authors: We thank the reviewer for the suggestions.

Regarding the radiological details to justify the LiRADs-4, an inhomogeneous margin enhancement directly after a segmental artery as well as the partial washout and the necrosis zone occurring in the course was detected. The findings are now presented in the axial plane spectral-CT-scan (**Fig 1**). We have added the description of the different phases and

rearranged **Fig 1**. Additionally, we have described the dates of the CT scans in the results section of the manuscript.

We have included this information in the revised manuscript as follows:

Results page 4 paragraph 1

A 74-year-old female patient with a history of continued alcohol abuse (one bottle of wine/ day for more than 10 years; 9 units/day) and malnutrition (BMI 12.8 kg/m²) presented in the emergency department with multiple fractures of the femur and pelvis after falling at home. Initial computed tomography (CT) incidentally revealed an 18x17 mm hypervascular nodule in liver segment 8 in November 2019 (**Fig. 1a**). The lesion showed complete wash-out in the venous/delayed phase and was therefore classified as LI-RADS 4 (probably HCC, biopsy recommended)⁷.

and

A follow-up thoracoabdominal CT scan after approximately 7 months (June 2020) showed an increase in nodular size up to 21x32 mm, with inhomogeneous arterial phase hyperenhancement and venous/delayed phase washout (**Fig. 1b**). There was a nodule-in-nodule pattern and a threshold growth of more than 50% within 6 months, both features supporting an upgrade to LI-RADS 5 (definitive HCC). Alpha-fetoprotein was not elevated at 4.7 kIU/l (normal range < 5.8 kIU/l) and showed no relevant increase over time. A new pulmonary focus was demarcated in the right upper lobe in November 2019, which was, however, assessed as indeterminate. The patient now undergoes regular CT scan controls. The last CT scan in September 2021 did not show any metastatic suspicion.

Figure updated: Fig. 1

Please highlight that GS is the lower panel in Fig 2a. Highlight the liver segment number might be helpful from a surgical perspective. If the same colours are to match between 3a and b should be a clearer match in tone, currently it is unclear what the colours add to the labels beneath in 3b.

Authors: We thank the reviewer for the suggestions, the figures have been updated as suggested.

Figures updated: Fig. 2

It is assumed that CT based chest staging was also performed. Can this be reported also please as evidence of standard clinical practice for exclusion of radiologically apparent metastasis?

Authors: We thank the reviewer for this important comment. In addition to the abdominal CT, a thoracic CT scan was performed as part of standard clinical practice. A new pulmonary focus was demarcated in the right upper lobe in November 2019, which was, however, assessed as indeterminate. The patient now undergoes regular CT scan controls. The last CT scan in September 2021 did not show any metastatic suspicion. We have added the information in the results part of the manuscript (**Results page 4 paragraph 1**).

The statement regarding male patients is irrelevant in this case (line 247) and can be removed.

Authors: We thank the reviewer for the suggestion. The sentence has been removed.

With reference to a mechanistic role of autophagy and HCC, Barthelet et al. (DOI: 10.1126/sciadv.abf9141) recently demonstrated a role for ATG5/7 loss in promoting HCC in a steatotic murine model of HCC. I would suggest also discussion related to this study in the associated paragraph discussed autophagy mediated carcinogenesis (line 175).

Authors: We thank the reviewer for this suggestion. In the revised version of the manuscript, we have now integrated the following paragraph in the discussion:

Discussion page 10 paragraph 1

Indeed, a recent study published by Barthelet et al.⁸ showed that loss of *ATG5* in the context of hemizygoty of *PTEN* in mice causes the development of HCC involving ductular reaction⁸. Mechanistically, the authors showed that loss of autophagy is followed by activation of YAP/TAZ in hepatocytes leading to its differentiation into biliary-like liver progenitor cells (ductular reaction) that ultimately lead to HCC.

Minor Typos

Line 51 "that present in HCC"

Line 68 "aminotransferase(s)"

The sentence commencing line 123 should be broken up to make it more readable.

Authors: We thank the reviewer for the corrections. In the revised version of the manuscript, all the highlighted issues have been corrected.

Reviewer #2 (Remarks to the Author):

This manuscript reported a 74-year-old female patient with an FNH and associated concomitant hepatocellular carcinoma. Immunophenotypic and exome sequencing showed that HCC may developed from FNH.

Major questions:

1) High grade dysplastic nodule and early HCC is very similar. Which evidences show that HCC2 is not dysplastic nodule ?

Authors: We acknowledge the concerns of the reviewer. Edmondson grade 1 early HCC and high-grade dysplastic nodule may be very similar and it can be very difficult to distinguish them from each other in some cases. We have amended the manuscript using the following nomenclature: Hepatocellular carcinoma number 2 was defined as Edmondson grade 1 HCC or as high-grade dysplastic nodule (HCC2/HGDN)

2) TERT alteration (promoter mutation, copy number variation, HBV insertion) is usually occurred in early progress of HCC. Whether authors observe TERT alteration? If whole exome sequencing can not cover the promoter region of TERT, expression of TERT should be tested by IHC or other approaches. Whether TERT has higher expression in HCC1 and HCC2 than in FNH ?

Authors: We thank the reviewer for the suggestion. Given that WES does not cover the promoter region of *TERT*, in the revised version of the manuscript we performed Sanger sequencing for the 2 hotspot mutations in the *TERT* promoter commonly found in HCC (-c.124C>T and -c.146C>T). We found that the FNH and the 2 associated lesions harbor the hotspot mutation. Moreover, given that somatic alteration in the *TERT* promoter may result in changes at the RNA level we have evaluated the *TERT* expression by QPCR in all the lesions and observed no difference in the 3 mutant samples. Additionally, we also investigated the possible gain/amplification at the genome level using WES. As reported in **Fig. 3c** we did not observe any copy number gain/amplification at the *TERT* locus (5p15.33).

We have now included this data in the results and material and methods sections including an additional supplementary figure (**Fig. S2**) as follow:

Results page 6 paragraph 2

Additionally, given that the WES does not cover the promoter region of *TERT*, we performed Sanger sequencing for the 2 hotspot mutations commonly found in HCC (-c.124C>T and -c.146C>T). We found that the FNH and the two HCC components harbor the hotspot mutation -c.124C>T. The effect of this mutation was also investigated at the *TERT* mRNA level. No difference in *TERT* expression was observed between the lesions (**Fig. S2**).

Materials and Methods page 13 paragraph 2

PCR amplification, Sanger sequencing and quantitative real-time PCR

For the identification of hotspot somatic mutations in *TERT* promoter, primer sets that amplify the hotspot sites of the *TERT* promoter were designed as previously described⁹ and are available in our previously published study¹⁰. PCR amplification was performed from 100 ng of genomic DNA using the AmpliTaq Gold 360 Master Mix Kit (Life Technologies) on a SimpliAmp Thermal Cycler (ThermoFisher) as previously described¹⁰. Sequencing was performed using purified PCR fragments (QIAquick PCR Purification Kit, Qiagen) on an ABI 3730 capillary sequencer using the ABI BigDye Terminator chemistry (v3.1, Life Technologies). Sequences of the forward and reverse strands were analyzed using 4Peaks (<https://nucleobytes.com/4peaks/>). All analyses were performed in triplicate.

RNA extraction from FFPE tissues was performed using RecoverAll Total Nucleic Acid Kit for FFPE (ThermoFisher) according to manufacturer's guidelines. Quantitative RT-PCR analysis was performed using SYBR Green. GAPDH was used as housekeeping genes for normalization. mRNA fold expression change was calculated by the $2^{-\Delta\Delta CT}$ method as previously described¹¹. The following Primers set were used: *GAPDH* Forward 5'-AGGTGAAGGTCGGAGTCAACG-3' and Reverse 5'-TGGAAGATGGTGTATGGGATTT-3' and *TERT*¹² Forward 5'-GCCGATTGTGAACATGGACTACG-3' Reverse 5'-GCTCGTAGT TGAGCACGCTGAA-3'.

Figure added: Fig. S2

Minor questions:

1) The raw data of exome sequencing should be submitted to a public database (such as NCBI SRA).

Authors: We apologize that this was in progress at the time of initial submission. In the revised manuscript we have added the following statement:

Data availability

Sequencing data are available on the European Genome-Phenome Archive database under the accession number EGAD00001007702.

Digital pathology images have been deposited on the Zenodo database under the accession number <https://doi.org/10.5281/zenodo.5554337>

Reviewer #3 (Remarks to the Author):

The authors present a case of concurrent hepatocellular carcinoma (HCC) and focal nodular hyperplasia (FNH) arising in non-cirrhotic liver. FNH is typically considered to be a non-neoplastic lesion formed in reaction to local vascular flow aberrations. The authors show a lesion which has 3 distinct histomorphologies, two of which are definitely HCC. They have evaluated the areas with a small panel of immunostains (glutamine synthetase, CD34, glypican-3), as well as reticulin. They went on to microdissect the separate areas and perform whole exome sequencing. They identified 80 mutations shared between the 3 areas. Clonality assay demonstrated multiple clones in each area. The authors conclude that their findings support the hypothesis that FNH can be a precursor to HCC.

Authors: We thank the reviewer for the favorable assessment of our work.

1. My major concern with the paper is in regard to the "FNH". Specifically, I am concerned that this may actually represent an inflammatory hepatocellular adenoma (which would be a known precursor lesion for HCC and can result in HCC in non-cirrhotic liver). These can very much resemble FNH, with presence of ductules, inflammation, and pseudoportal tracts (even sometimes septa); in fact, before they were discovered to be adenomas, they were called "telangiectatic FNH". Based on the photomicrographs provided and the glutamine synthetase stains shown (I am not convinced of a map-like/geographic pattern -- and HCAs can have glutamine synthetase staining, including perivenular), I think more clarification is required. Immunostains for CRP and SAA should be performed. Mutational analysis, if not already performed, for mutations in IL6ST, GNAS, STAT3, and CTNNB1 should be completed (if already completed, the findings should be specifically described).

Authors: we thank the reviewer for this observation and we understand the concerns raised. The lesions described in this manuscript have been histologically characterized independently by six pathologists (C.E., J.V., A.T., M.S.M., L.D.T., and L.M.T.) and all of them agreed on the results presented in this study. However, following the reviewer's suggestion, we have performed immunostain for SAA and CRP. An analysis of SAA and CRP in our FNH revealed negativity for SAA and partial positivity of CRP. CRP has been reported to be positive in 78% of FNH¹³ whereas SAA is positive in 92.6% of inflammatory hepatocellular adenoma and has higher specificity than CRP for the diagnosis of inflammatory hepatocellular adenomas¹³. Our results, therefore, suggest that our FNH lesion is unlikely to be an inflammatory HCA.

In regards to the glutamine synthetase immunostaining, we have replaced the old micrograph with a new one in **Fig 2** (we kept the old one in **Fig S1**) to better show the staining pattern.

Additionally, we have provided digital images for all the staining performed (<https://doi.org/10.5281/zenodo.5554337>, and the specific GS stainings can be found GS-1 and GS-2)

In regard to the mutational analysis, as reported in **Fig. 3b/ Supplementary Table 1**, no somatic alterations in *IL6ST*, *GNAS*, *STAT3*, and *CTNNB1* were identified in these lesions.

We have added these new results in the revised version of the manuscript as follow:

Results: the paragraph “Pathologic characterization of the lesions” on pages 5 and 6 and “Genomic characterization reveals clonal evolution of the FNH to HCC” on page 6 and 7 have been updated.

Material and Methods (Immunophenotypic characterization page 11): Sequential 3 µm-thick sections of formalin-fixed, paraffin-embedded (FFPE) tumoral tissue were used. Deparaffinized serial sections were stained by Hematoxylin and Eosin (H&E) and Novotny reticulin stain. Histopathologic HCC grading was performed according to the Edmondson system. Immunohistochemical staining were performed with monoclonal antibodies against glutamine synthetase (clone GS-6, mouse, Ventana/Roche, Mannheim, Germany), Glypican-3 (clone 1G12, mouse, Ventana/Roche, Mannheim, Germany), CD34 (Ventana/Roche, Mannheim, Germany), Ki67 (clone Mib1, catalog no IR626, Dako, Carpinteria, CA, USA), Serum Amyloid-A (clone Mc1, catalog no IR605, Dako, Carpinteria, CA, USA) and C Reactive Protein (clone ab32412, Abcam, Cambridge, MA, USA) on a Benchmark immunostainer (Ventana, Roche) according to the manufacturers' instructions.

Discussion page 8 paragraph 3

The histological view of FNH nodule consists of numerous foci of hepatocytes intersected with arteria and bile ductuli-rich fibrous bands, which is diagnostic for FNH together with the distinctive “map-like” patchy GS expression of the hepatocytes. Given the possible morphological similarity of FNH with inflammatory type hepatocellular adenoma (IHA) in some cases, we performed additional immunostains (SAA and CPR) to rule out this alternative diagnosis. The immunostaining revealed negativity for SAA and partial positivity for CPR. A study performed by Joseph et. al. has been shown that the SAA expression is positive in the vast majority of IHA (92.6%) while the CRP expression was found in 78% of FNH¹³. The diagnosis of IHA was then excluded based on these results together with other morphological features. The diagnosis of IHA was then excluded based on these results together with other morphological features. These observations were further supported at the genomic level, given the absence of common driver genetic somatic alterations usually found IHA such as mutations in *CTNNB1*, *IL6ST*, *GNAS*, *STAT3*¹⁴.

2. The adjacent non-lesional liver should also be addressed. It would be of particular interest to perform similar mutational analysis on the non-lesional liver right next to the tumour, and then more remotely. If similar mutations are seen in the background liver and the FNH-like area, then the support this as a precursor lesion is lessened; however, if no mutations are found in the background, then the conclusion (assuming I-HCA has been ruled out) is supported.

Authors: we apologize for the lack of clarity in regards to this point, we have used the non-tumoral liver as germline control to investigate the somatic changes in the lesions. The investigation of germline polymorphisms that may be implicated in the pathogenesis of FNH is

out of the scope of this manuscript. Additionally, the analysis of germline polymorphisms from a single case may raise more questions than answers.

3. The authors should consider review and discussion of peritumoural hyperplasia, which is a FNH-like lesion. If they feel their FNH-like area is not peritumoural hyperplasia (and, if proven not to be inflammatory hepatocellular adenoma), they should make a statement defending their decision.

The authors may find this article helpful: Arnason T, Fleming KE, Wanless IR. Peritumoural hyperplasia of the liver: a response to portal vein invasion by hypervascular neoplasms. *Histopathology*. 2013;62:458-64.

Authors

Authors: we thank the reviewer for the suggestion. We agree that peritumoural hyperplasia may resemble FNH¹⁵. PTH is defined as a rim of hepatocytes that surrounds the circumference of the tumor in a cuffing manner, while the FNH in our case is located on one side of the HCC and has a nodular morphology. (Fig 2b)

In the revised version of the manuscript, we have discussed this point in the discussion

Discussion page 9 paragraph 2

Peritumoural hyperplasia (PTH) is another entity that can resemble FNH. Arnason et al. described PTH as a hyperplastic response to increased blood flow in the peritumoural parenchyma. Its characteristic morphology is a rim of hepatocytes surrounding the circumference of HCC like a cuff¹⁵. In our case, the lesion was a nodular lesion localized at the neighbor of the tumor, instead of encircling the HCC. Thus, PTH was not considered for the diagnosis.

Overall, I think more work needs to be performed to support the conclusion and strengthen the paper. If the authors can further support that the FNH-like area truly is FNH and that the background liver harbours no significant mutations, then this may be paradigm changing (a lesion thought to be non-neoplastic and not a precursor may indeed be!) and affect patient care (perhaps more resections, more radiographic follow-up).

Additional References

1. European Association for the Study of the Liver (EASL). EASL Clinical Practice Guidelines on the management of benign liver tumours. *J. Hepatol.* **65**, 386–398 (2016).
2. Fujimoto, A. *et al.* Whole-genome mutational landscape and characterization of noncoding and structural mutations in liver cancer. *Nat. Genet.* **48**, 500–509 (2016).
3. Letouzé, E. *et al.* Mutational signatures reveal the dynamic interplay of risk factors and cellular processes during liver tumorigenesis. *Nat. Commun.* **8**, 1315 (2017).
4. Rosenthal, R., McGranahan, N., Herrero, J., Taylor, B. S. & Swanton, C.

- DeconstructSigs: delineating mutational processes in single tumors distinguishes DNA repair deficiencies and patterns of carcinoma evolution. *Genome Biol.* **17**, 31 (2016).
5. Nik-Zainal, S. & Morganella, S. Mutational Signatures in Breast Cancer: The Problem at the DNA Level. *Clin. Cancer Res.* **23**, 2617–2629 (2017).
 6. Popova, T. *et al.* Ploidy and large-scale genomic instability consistently identify basal-like breast carcinomas with BRCA1/2 inactivation. *Cancer Res.* **72**, 5454–5462 (2012).
 7. Mitchell, D. G., Bruix, J., Sherman, M. & Sirlin, C. B. LI-RADS (Liver Imaging Reporting and Data System): summary, discussion, and consensus of the LI-RADS Management Working Group and future directions. *Hepatology* **61**, 1056–1065 (2015).
 8. Barthelet, V. J. A. *et al.* Autophagy suppresses the formation of hepatocyte-derived cancer-initiating ductular progenitor cells in the liver. *Sci Adv* **7**, (2021).
 9. Weinreb, I. *et al.* Hotspot activating PRKD1 somatic mutations in polymorphous low-grade adenocarcinomas of the salivary glands. *Nat. Genet.* **46**, 1166–1169 (2014).
 10. Piscuoglio, S. *et al.* Massively parallel sequencing of phyllodes tumours of the breast reveals actionable mutations, and TERT promoter hotspot mutations and TERT gene amplification as likely drivers of progression. *J. Pathol.* **238**, 508–518 (2016).
 11. Livak, K. J. & Schmittgen, T. D. Analysis of Relative Gene Expression Data Using Real-Time Quantitative PCR and the $2^{-\Delta\Delta CT}$ Method. *Methods* vol. 25 402–408 (2001).
 12. Gong, C. *et al.* hTERT Promotes CRC Proliferation and Migration by Recruiting YBX1 to Increase NRF2 Expression. *Front Cell Dev Biol* **9**, 658101 (2021).
 13. Joseph, N. M. *et al.* Diagnostic utility and limitations of glutamine synthetase and serum amyloid-associated protein immunohistochemistry in the distinction of focal nodular hyperplasia and inflammatory hepatocellular adenoma. *Mod. Pathol.* **27**, 62–72 (2014).
 14. Védie, A.-L., Sutter, O., Ziol, M. & Nault, J.-C. Molecular classification of hepatocellular adenomas: impact on clinical practice. *Hepat Oncol* **5**, HEP04 (2018).
 15. Arnason, T., Fleming, K. E. & Wanless, I. R. Peritumoral hyperplasia of the liver: a response to portal vein invasion by hypervascular neoplasms. *Histopathology* **62**, 458–

464 (2013).

Reviewers' comments:

Reviewer #1 (Remarks to the Author):

I thank the author's for their time and effort in the revision of the manuscript. I find the additional discussion they provide in their response to the reviewers comments reassuring across a number of levels and the article itself to have been significantly improved by the revision process.

The additional clinical description, pathological assessment and molecular classification of the lesions is particularly appreciated. I also find the figures improved in their clarity.

Overall I am satisfied with the revision.

My only minor comments are:

That the labels within panels of Figure 2 are difficult to make out and that ideally panel A and B within the figure should be split for clarity.

In the Fig S1 legend "(i) GS..." should read (I) GS...

The insert within panel L is not helpful and ideally a scale bar should be provided and the scales should be legible throughout.

Reviewer #2 (Remarks to the Author):

Most of my original questions have been addressed.

Fig. S2b shows there is no difference in TERT expression between the lesions. Since both FNH and HCC have TERT mutation, it's better to compare TERT expression between normal (control) tissue with mutated tissues to see the effect of mutation on expression.

Reviewer #3 (Remarks to the Author):

I thank the authors for their thoughtful and thorough responses to the reviewers' questions and concerns. I think that the paper is much stronger now.

My major concern previously was that the FNH portion of the lesion actually represented an I-HCA. With further pathologic review, immunostains, and mutational analysis, the authors have convinced me that it is not I-HCA. As well, they have discussed why it is not compatible with peritumoural hyperplasia.

At this point, my only strong suggestion is that the authors also discuss the possibility that the FNH did not occur first, but that the hepatocellular neoplasia did. HGDN and HCC are arterialized lesions, as shown, and can disrupt local vascular flow, which could lead to FNH. Indeed, this is the commonly accepted hypothesis for why FNH occurs next to lesions such as HCA and HCC. The authors should consider that the HGDN/HCC induced FNH and that the shared mutations are due to the FNH forming in an area of hepatocellular parenchyma that already had those mutations, as a precursor to HCC (such as small cell change).

Considering the above, I would suggest including a paragraph discussing this possibility in the Discussion, as well as editing Discussion lines 193-195 to reflect that the authors think this (FNH  HCC) is what occurred, perhaps "...at diagnosis and we think is likely to have progressed..." and discussion lines 250-255, perhaps "FNH can share some similar genomic alterations with HCC and may not be absolutely benign and may rarely progress..."

I do very much appreciate the authors acknowledgement that this is one case and that more research is needed, rather than trying to make recommendations for follow-up or surveillance at this time.

Figure 2 c and e; supplementary figure 1l - I do not think that the glutamine synthetase portions need higher power insets -- in fact, I think the insets in the GS portions obscure the lower power photos which are more important here. Suggest removing the insets for the GS photos.

Terminology change suggested -- line 119 and supplementary figure legend 1. Suggest using "capillarization" and/or "arterialization", the typical terminology for CD34+ sinusoidal endothelial cells in the liver, instead of "endothelialization".

If the above can be done, particularly the addition to the discussion, then I think this paper can be accepted for publication.

We are delighted that the Reviewers and the Editorial Board of Communication Medicine found our revised manuscript improved from the previous version and we are pleased that our additional analyses have convinced the reviewers.

In this revised version (R2) we have addressed all the minor points raised.

Below you can find a point-by-point response to the reviewers' comments:

Reviewers' comments:

Reviewer #1 (Remarks to the Author):

I thank the author's for their time and effort in the revision of the manuscript. I find the additional discussion they provide in their response to the reviewers' comments reassuring across a number of levels and the article itself to have been significantly improved by the revision process.

The additional clinical description, pathological assessment and molecular classification of the lesions is particularly appreciated. I also find the figures improved in their clarity.

Overall I am satisfied with the revision.

Authors: We thank the reviewer for the favorable assessment of our revised work.

My only minor comments are:

That the labels within panels of Figure 2 are difficult to make out and that ideally panel A and B within the figure should be split for clarity.

Authors: We thank the reviewer for the suggestion. In the revised version of the manuscript the labels within panels of Figure 2 have been increased in size and panel A and B have been clearly separated.

In the Fig S1 legend "(i) GS..." should read "(I) GS..."

The insert within panel L is not helpful and ideally a scale bar should be provided and the scales should be legible throughout.

Authors: We thank the reviewer for the suggestion. In the revised version of the manuscript the legend of Figure S1 has been corrected, the scale bars have been adjusted and the insert within panel L has been removed.

Reviewer #2 (Remarks to the Author):

Most of my original questions have been addressed.

Authors: We thank the reviewer for the favorable assessment of our revised work.

Fig. S2b shows there is no difference in TERT expression between the lesions. Since both FNH and HCC have TERT mutation, it's better to compare TERT expression between normal (control) tissue with mutated tissues to see the effect of mutation on expression.

Authors: We thank the reviewer for the suggestion. We have now added *TERT* expression in the non-tumoral liver to the figure and modified the text as follows:

Results page 7-8 paragraph 2

We found that the FNH and the two HCC components harbor the hotspot mutation –c.124C>T. The effect of this mutation was also investigated at the TERT mRNA level. *TERT* expression was higher in the lesions compared to the matched non-tumoral liver (Fig. S2).

Reviewer #3 (Remarks to the Author):

I thank the authors for their thoughtful and thorough responses to the reviewers' questions and concerns. I think that the paper is much stronger now.

My major concern previously was that the FNH portion of the lesion actually represented an I-HCA. With further pathologic review, immunostains, and mutational analysis, the authors have convinced me that it is not I-HCA. As well, they have discussed why it is not compatible with peritumoural hyperplasia.

Authors: We thank the reviewer for the favorable assessment of our revised work.

At this point, my only strong suggestion is that the authors also discuss the possibility that the FNH did not occur first, but that the hepatocellular neoplasia did. HGDN and HCC are arterIALIZED lesions, as shown, and can disrupt local vascular flow, which could lead to FNH. Indeed, this is the commonly accepted hypothesis for why FNH occurs next to lesions such as HCA and HCC. The authors should consider that the HGDN/HCC induced FNH and that the shared mutations are due to the FNH forming in an area of hepatocellular parenchyma that already had those mutations, as a precursor to HCC (such as small cell change).

Considering the above, I would suggest including a paragraph discussing this possibility in the Discussion, as well as editing Discussion lines 193-195 to reflect that the authors think this (FNH  HCC) is what occurred, perhaps "...at diagnosis and we think is likely to have progressed..." and discussion lines 250-255, perhaps "FNH can share some similar genomic alterations with HCC and may not be absolutely benign and may rarely progress..."

Authors: We thank the reviewer for the suggestion. We amended/added the following sentences in the discussion:

Page 9 Paragraph 2

Here we performed a genetic analysis of one FNH with two associated lesions classified as Edmondson grade I HCC or high-grade dysplasia and an Edmondson grade II HCC components and found that the FNH is composed of multiple clones at diagnosis. *We think our results suggest the FNH likely* progressed to HCC through clonal selection and/or the acquisition of additional genetic events.

Page 12 Paragraph 1

While several studies have reported co-occurrence of FNH and HCC^{4-6,9}, to our knowledge this is the first study showing a clonal relationship between these lesions. Moreover, we found uncommon genetic alterations in the HCC that might be associated with hepatocarcinogenesis in the background underlying FNH. Importantly, our results suggest in extremely rare cases, FNH can share some similar genomic alteration with HCC, *may not be*

absolutely benign and may, albeit rarely, progress to HCC. Although our results suggest that FNH was a non-obligate precursor lesion of HCC, another hypothesis is that given that HGDN and HCC are arterIALIZED lesions, the disrupted local vascular flow may lead to the development of FNH. We cannot fully exclude the possibility that the FNH developed in an area of an HCC precursor that already harbored the genetic alterations. However, given the clonal frequencies of the observed mutations, we think it is more likely that the FNH progressed to HCC through clonal selection.

I do very much appreciate the authors acknowledgement that this is one case and that more research is needed, rather than trying to make recommendations for follow-up or surveillance at this time.

Figure 2 c and e; supplementary figure 11 - I do not think that the glutamine synthetase portions need higher power insets -- in fact, I think the insets in the GS portions obscure the lower power photos which are more important here. Suggest removing the insets for the GS photos.

Authors: We thank the reviewer for the suggestion. In the revised version of the manuscript the higher power insets for GS have been removed.

Terminology change suggested -- line 119 and supplementary figure legend 1. Suggest using "capillarization" and/or "arterialization", the typical terminology for CD34+ sinusoidal endothelial cells in the liver, instead of "endothelialization".

Authors: We thank the reviewer for the suggestion. In the revised version of the manuscript, we have changed the terminology as suggested.

If the above can be done, particularly the addition to the discussion, then I think this paper can be accepted for publication.

REVIEWERS' COMMENTS:

Reviewer #2 (Remarks to the Author):

I think this paper can be accepted for publication.